# A Comprehensive Review of Natural Compounds for Wound Healing: Targeting Bioactivity Perspective

**DOI:** 10.3390/ijms23179573

**Published:** 2022-08-24

**Authors:** Xuan-Tung Trinh, Nguyen-Van Long, Le Thi Van Anh, Pham Thi Nga, Nguyen Ngan Giang, Pham Ngoc Chien, Sun-Young Nam, Chan-Yeong Heo

**Affiliations:** 1Department of Plastic and Reconstructive Surgery, Seoul National University Bundang Hospital, Seongnam 13620, Korea; 2Department of Medical Device Development, College of Medicine, Seoul National University, Seoul 03080, Korea; 3Department of Plastic and Reconstructive Surgery, College of Medicine, Seoul National University, Seoul 03080, Korea

**Keywords:** wound healing, natural compounds, bioactivity, anti-inflammation, anti-oxidant, anti-bacterial, collagen promotion, targeting phase

## Abstract

Wound healing is a recovering process of damaged tissues by replacing dysfunctional injured cellular structures. Natural compounds for wound treatment have been widely used for centuries. Numerous published works provided reviews of natural compounds for wound healing applications, which separated the approaches based on different categories such as characteristics, bioactivities, and modes of action. However, current studies provide reviews of natural compounds that originated from only plants or animals. In this work, we provide a comprehensive review of natural compounds sourced from both plants and animals that target the different bioactivities of healing to promote wound resolution. The compounds were classified into four main groups (i.e., anti-inflammation, anti-oxidant, anti-bacterial, and collagen promotion), mostly studied in current literature from 1992 to 2022. Those compounds are listed in tables for readers to search for their origin, bioactivity, and targeting phases in wound healing. We also reviewed the trend in using natural compounds for wound healing.

## 1. Introduction

Wounds occur as a result of accidental or surgical trauma and from a variety of medical conditions. This wound often causes pain, inflammation, and loss of function, which affects a patient’s life and financial costs [1]. Wounds are classified as acute wounds or chronic wounds. Wound healing is a complex process of replacing damaged and dysfunctional cellular structures and tissue layers [2]. Acute wounds go through stages of healing, and signs of healing are well-defined within four weeks. Chronic wounds do not undergo normal progression through the healing phases, and healing is not apparent within four weeks. It can be said that the wound healing process depends on factors at the wound site, systemic mediators, type of injury, or any underlying disease [3]. Wound treatment is mainly performed by strategies such as physical closure of the wound margin, sutures, and dressings. When the wound is inaccessible, leave the wound open and let the damaged area clear itself and fill with connective tissue, and the healing process occurs sequentially through phases.

Natural compounds have been used for thousands of years to treat wounds. Natural compounds are found in many plants and animals, which are an abundantly available source for wound treatment. They have proven effective in healing through Chinese and Indian traditional medicines. Due to a vast number of natural compounds, reviews of those compounds would benefit readers and researchers in systematically finding interesting compounds and developing new products for wound healing treatment. Previously, many review papers discussed natural compounds for wound healing treatment [1,4,5,6,7,8,9,10,11,12]. For example, Ryall and colleagues discussed current advancements in skin delivery of natural bioactive compounds for wound management (e.g., turmeric, green tea, honey, garlic, aloe vera, etc.) [4]. Vitale et al. focused on medicinal plants’ phytochemistry and biological activity in wound healing [5]. Ataide and colleagues discussed the activities of pro-wound healing compounds and their mode of action [7]. Dumitru et al. discussed bee products for wound healing treatment [13]. Fana et al. reviewed natural wound healing compounds in traditional Iranian medicine [11]. Those reviews provided many natural compounds for wound healing treatment. However, they only gave tables or lists of natural compounds regarding categories, bioactivities, and mode of action. Those reviews lack discussion on which phase of wound healing natural compounds are affected. Readers might find difficulty when they want to search for information on interesting compounds (wound healing phase, category, chemical formula, mechanism, etc.).

Herein, we give a review of natural compounds (from both plants and animals) that play important roles in wound healing, also their healing mechanisms and limitations in use. We classified those compounds based on targeting bioactivities for wound healing. We also summarized the current trends in using natural compounds. We provided data tables for readers to search natural compounds regarding their origin, bioactivity, and targeting phases in wound healing.

## 2. The Process of Wound Healing

Wound healing is a process consisting of four phases: hemostasis, inflammation, proliferation, and remodeling. Illustration of the wound healing process is shown in Figure 1.

### 2.1. Hemostasis Phase

Wound healing first begins with hemostasis. The lymphatic vessels are injured in this phase, and blood flows out to remove microorganisms and antigens [14]. The body will activate different clotting cascades and thrombocytes to agglomerate by exposed collagen. At the same time, platelets activate vasoconstriction to reduce blood loss and fill tissue gaps in injured vessels with blood clots containing cytokines and growth factors [15]. The clot contains the molecules fibrin, fibronectin, vitronectin, and thrombospondin, which form a temporary matrix as a scaffolding structure for the migration of leukocytes, keratinocytes, fibroblasts, and endothelial cells, and it is a reservoir of growth factors that stabilize blood clots and avoid bleeding.

### 2.2. Inflammation Phase

The second phase of wound healing is inflammation which focuses on cleaning the wound and preparing for new tissue formation in the wound. This stage has the appearance of neutrophils and lasts about 2–5 days from when the wound becomes infected. Neutrophils can phagocytize and secrete proteases (elastase, cathepsin G, proteinase 3) that help destroy bacteria in the wound and deco remove debris. Neutrophils also release mediators (TNF-α, IL-1 and IL-6) to amplify the inflammatory response, stimulating VEGF and IL-8 to respond to repair during wound healing [16]. The macrophage process then supports the ongoing process by phagocytosis of the debris and secretion of growth factors, chemokines, and cytokines [17]. Macrophages promote and address inflammation, eliminate apoptotic, and support cell proliferation and tissue recovery after injury [18]. In the inflammatory phase, there are often symptoms of edema, erythema and pain.

### 2.3. Proliferation Phase

The proliferation phase is the most important phase of the wound healing process and lasts from 6 to 21 days. During the proliferation phase of wound healing, the wound is healed with fresh collagen and extracellular matrix tissue. After that, the wound shrinks as new tissues develop. A new network of blood vessels must be created for granulation tissues to remain healthy and receive an adequate supply of nutrients and oxygen. The modulation of fibroblasts toward myofibroblasts promotes the formation of granulation tissue. The myofibroblasts are characterized by the capacity to produce force and synthesize extracellular matrix components that allow the contraction of granulation tissue [19]. By gripping the wound boundaries and pulling them together, myofibroblasts use a technique akin to that of smooth muscle cells to close the wound. In the initial stages of wound healing, granulation tissue appears pink or red and has an uneven texture. Furthermore, healthy granulation tissue is clot-resistant [20,21]. Dark granulation tissue may be brought on by an infection, ischemia, or insufficient perfusion. Near the conclusion of the proliferation phase, epithelial cells resurface the wound. Keeping wounds moist accelerates epithelialization. Epithelialization occurs when occlusive or semi-occlusive dressings are applied within 48 h after the injury. This is because adequate tissue humidity is maintained. One accomplishment of the proliferation phase is replacing the temporary fibrin matrix with a new matrix made of collagen fibers, proteoglycans, and fibronectin to restore the structure and function of tissues. Another crucial stage of healing is angiogenesis, or the ingrowth of new capillaries to replace previously damaged vessels and restore circulation. The creation of granulation tissue and epithelialization are other important phenomena in this healing period. In the proliferation phase of healing, fibroblasts are the most important cells [22,23]. For fibroblasts to migrate in the extracellular matrix, they must first recognize and interact with particular matrix components. Fibroblasts in the normal dermis are usually dormant and sparsely scattered, but they are active and plentiful in the provisional matrix wound site and granulation tissue [24,25]. Their migration and aggregation in the wound site necessitate morphological changes and the production and secretion of proteases to clear a passage from the ECM into the wound site. The chemotactic growth factors, cytokines, and chemokines concentration gradient, as well as the alignment of the fibrils in the ECM and provisional matrix, control the direction of fibroblast migration. Rather than crossing these fibrils, fibroblasts prefer to move along them [26,27]. To help them move through the matrix, fibroblasts produce proteolytic enzymes on a local level. Collagenase (MMP-1), gelatinases (MMP-2 and MMP-9) that destroy gelatin substrates, and stromelysin (MMP-3), which has various protein substrates in the ECM, are three kinds of MMPs released by fibroblasts [28,29]. After migrating into the matrix, fibroblasts change shape, settle down, and begin to proliferate and generate granulation tissue components such as collagen, elastin, and proteoglycans. Fibroblasts connect to the provisional fibrin matrix cables and begin producing collagen [19,30]. Type III collagen, like other extracellular matrix proteins and proteoglycans, is generated in high amounts at first [31]. Collagen mRNA is connected to polyribosomes on the endoplasmic reticulum, where new collagen chains are formed after transcription and processing. A crucial stage in this process involves proline and lysine residue hydroxylation.

### 2.4. Remodeling Phase

Closure of acute and chronic wounds is regarded as the wound healing endpoint in most clinical settings, yet wounds can continue to undergo remodeling or tissue maturation for months or even years [32,33]. This final stage of wound healing decides whether scarring will occur and whether the wound will reoccur. Regression of the neo vasculature, a periodic deposition to the ECM, and subsequent reconstruction of granulation tissue to scar tissue are all part of the remodeling phase [26]. Collagen III makes up the majority of granulation tissue, which is gradually replaced by the stronger collagen I as the wound heals. This occurs due to simultaneous collagen I production and collagen III lysis, followed by ECM remodeling [34]. In the remodeling phase, scar tissues are created, and it might take several months or years to complete, depending on the severity and location of the wound, and used therapeutic procedures. During this time, the new tissue gradually gets stronger and more flexible. Elasticity and tensile strength of the skin are both getting stronger because of collagen synthesis. After re-epithelialization, macrophages regain their phagocytic phenotype. Excessed cells and matrix no longer required for wound healing are phagocytosed by Mreg or M2c-like macrophages [24].

## 3. Classification of Natural Compounds for Wound Healing by Their Properties

From the literature search, we collected a list of 137 research articles [30,35,36,37,38,39,40,41,42,43,44,45,46,47,48,49,50,51,52,53,54,55,56,57,58,59,60,61,62,63,64,65,66,67,68,69,70,71,72,73,74,75,76,77,78,79,80,81,82,83,84,85,86,87,88,89,90,91,92,93,94,95,96,97,98,99,100,101,102,103,104,105,106,107,108,109,110,111,112,113,114,115,116,117,118,119,120,121,122,123,124,125,126,127,128,129,130,131,132,133,134,135,136,137,138,139,140,141,142,143,144,145,146,147,148,149,150,151,152,153,154,155,156,157,158,159,160,161,162,163,164,165,166,167,168,169,170] relating natural compounds for wound healing. We classified them into groups regarding their bioactivities (i.e., anti-inflammation, anti-oxidant, antibacterial, collagen promotion, etc.) (Figure 2). The origin of those compounds (i.e., plant and animal) was also considered. Among bioactivities, anti-inflammation, anti-oxidant, anti-bacterial, and collagen promotion are studied the most. Therefore, in this study, we focused our discussion on natural compounds regarding these bioactivities. A data table containing a list of those literature and the compounds were provided in the Appendix A.

### 3.1. Natural Compounds with Anti-Inflammation Properties 

The inflammatory response is an important process in wound healing. Inflammation and anti-inflammation affect the process of hemostasis, removal of harmful microorganisms, damaged tissues, and wound cleaning [171]. However, if the inflammation phase is prolonged, it will lead to a pathological condition and affect the wound healing process [172,173]. To solve this problem, compounds with an anti-inflammation activity that impact the wound healing process are a therapeutic target. A list of compounds reading anti-inflammation is shown in Table 1.

#### 3.1.1. Myricetin

Myricetin (Myr) is a flavonoid that has been reported for wound healing [47,59]. Myricetin is present in many fruits and has many biochemical properties such as antioxidant, anti-allergic, anti-inflammation, and immunomodulatory function [178,179,180,181]. Elshamy et al. isolated myricetin from *Tecomaria capensis v*. *aurea* and examined its wound healing ability in albino rats [47]. Myr affects inflammatory cytokines such as tumor necrosis factor-α (TNF-α), cluster of differentiation 68 (CD68), as well as interleukin-1β (IL-1β). Myr also showed increased expression of serum proinflammatory cytokines (e.g., IL-1β and TNF-α) and decreased expression of macrophage CD68. The above findings suggest that Myr could be used therapeutically in wound healing by enhancing inflammatory cytokines and systemic reorganization. Other than that, Sklenarova et al. investigated Myr’s ability to heal wounds [59]. This study showed the inhibition of proinflammatory cytokines production (e.g., IL-6 and IL-8) in skin cells by Myr.

However, myricetin is very poorly soluble in water [182]. This affects its bioavailability [183]. This limitation needs to solve by combining Myr with other compounds or biomaterial to improve the water solubility of Myr.

#### 3.1.2. Calophyllolide (CP)

Calophyllolide (CP) is isolated from *Calophyllum inophyllum* Linn and has been reported with anti-inflammatory, anti-microbial, and anti-coagulant activities [184,185,186]. Nguyen et al. studied the long-lasting anti-inflammatory effects of CP in the healing process [55]. They showed that CP treatment suppresses prolonged inflammation by downregulating IL-1β, IL-6, TNF-α, and upregulating IL-10. Moreover, CP inhibits MPO activity and increases M2 macrophage bias through upregulating M2-associated gene expression, leading to benefits in wound healing.

#### 3.1.3. Steroidal Glycoside

Steroidal glycoside is extracted from *Lilium longiflorum* Thunb. Di et al. confirmed that wound treatment with steroidal glycosides would upregulate early inflammatory genes such as IL2, IL4, IL10, CD40LG, IFNG, and CXCL11, remodeling genes like CTSG, F13A1, FGA, MMP and PLG) [56]. Concurrently, wound treatment with steroidal glycosides also displayed a selective downregulation of genes regarding inflammation (CXCL2 and CCL7) and regeneration (MMP7 and PLAT) [56]. The above findings suggest an impact of wound treatment with steroidal glycosides on wound healing, leading to early termination of the inflammatory response and shortening the early stages of tissue regeneration.

#### 3.1.4. Verbascoside (Acteoside)

Verbascoside is a phenolic compound with various bio-properties such as anti-inflammation, antioxidant, and healing [187,188,189]. Nathalia et al. isolated verbascoside from *Plantago australis* and examined its wound healing and anti-inflammatory activity [177]. This study has confirmed that verbascoside significantly reduced inflammatory cytokines (TNFα, INFγ, IL-6, MCP-1 and IL-12p70). In another study, Yasin et al. extracted verbascoside from *Plantago subulata* and evaluated its biological activity [62]. The in vitro test with RAW264.7 cell showed that when the cell was treated with verbascoside, the level of NO, PGE2, and TNF-α cytokines decreased. Another part of the study also confirmed that verbascoside from *Plantago subulata* has wound healing activities. The above studies suggested verbascoside has wound healing activities and may have related to anti-inflammation.

#### 3.1.5. Lupeol

Lupeol is a bioactive compound mainly found in *Bowdichia virgilioides* and fruit such as mango, soybean, and olive [58]. Researchers reported that Lupeol had antioxidant, antiinflammation, and antidiabetic activity [190,191,192]. To evaluate the wound healing ability of lupeol, Fernando et al. conducted experiments on rats and showed interesting results [176]. The results showed that lupeol effectively reduced inflammatory cytokines (e.g., NF-κB and IL-6) while increasing IL-10. Moreover, Lupeol also has effects on angiogenesis and cell proliferation by decreased expression of Vegf-A and increased expression of Hif-1α. There are markers for the angiogenic process and proliferation of wound healing. Another study was also done by Fernando et al., once again further identifying the wound healing activities of lupeol in the cream form [58]. The results showed that wound treatment with lupeol cream affects proinflammatory cytokines, such as reducing the expression of TNF-α, IL-1β and IL-6 and increasing the expression of IL-10 (Figure 3). In addition, lupeol treatment was also shown to improve vascular endothelial growth factor (VEGF) and epidermal growth factor (EGF) and increase gene expression of transforming growth factor beta-1 (TGF-β1) after 7 days. These are the factors that involve the proliferative phase in wound healing. Lupeol accelerates remodeling by increasing collagen fiber synthesis. These are studies that demonstrate the wound healing capacity of lupeol.

#### 3.1.6. Bilirubin

Bilirubin is a red-orange compound that is the end product of heme catabolism in mammals and also plays an important role in protecting cellular [193]. By speculating that Bilirubin might benefit wound healing, Azad et al. evaluated the wound healing process in rat skin when treated with bilirubin [54]. When the wound was treated with Bilirubin, pro-inflammatory factors (e.g., ICAM-1 and TNF-α) decreased, and interleukin-10 (IL-10) expression was increased. Wound contraction, hydroxyproline, and glucosamine levels were also increased in treated rats. In addition, Mahendra et al. also studied the effect of Bilirubin on growth factors, cytokines, and angiogenesis during wound healing in diabetic rats [138]. This study showed that pro-inflammatory cytokines such as TNF-α, MMP-9, and IL-1β decreased mRNA expression while increasing IL-10 expression. Gene expression of anti-oxidative, angiogenic agents (e.g., VEGF, HIF-1α, SDF-1α, TGF-β) was also upregulated in Bilirubin-treated rats. Wound closure, collagen deposition, and blood vessel formation in treated rats were also higher than in the control group (Figure 4). These results partly confirmed the role of Bilirubin in regulating pro-inflammatory and angiogenic factors in the wound healing process.

#### 3.1.7. Pinocembrin

Pinocembrin (5,7-dihydroxyflavonone) is one of the flavonoid compounds found in propolis, honey, and plants of the *Piperaceae* family [194]. The compound showed various potential bioactivities for healing treatment (e.g., anti-bacteria, anti-inflammation, anti-fibrosis, anti-oxidation) [194]. For example, Drewes and colleagues showed that pinocembrin had notable antibacterial activity toward *Staphylococcus aureus* (minimum inhibitory concentration of 6.3 μg/mL) and *Pseudomonas aeruginosa* (minimum inhibitory concentration of 45–63 μg/mL) [195]. Pinocembrin also showed anti-inflammatory activity against sheep red blood cell-induced delayed-type hypersensitivity reaction [196]. Pinocembrin is currently in traditional Chinese medicine for wound healing [103]. Li and colleagues investigated the effects of pinocembrin on skin fibrosis by in vitro and in vivo approaches [103]. The study showed that pinocembrin could significantly reduce bleomycin-induced skin fibrosis and fibrosis-related protein expression of keloid tissues in xenograft mice. They also confirmed the mechanism of anti-fibrotic activity of pinocembrin that pinocembrin suppressed TGF-β1/Smad signaling and attenuated TGF-β1-induced activation of skin fibroblasts.

### 3.2. Natural Compounds with Anti-Oxidant Properties

Antioxidants are one of the therapeutic targets to improve wound healing mechanisms, especially free radicals and oxidative reactions. They are known as an important factor in the regulation of the healing process [54,197,198]. A high concentration of oxidants in the wound inadvertently harms the wound and some enzymatic reactions during the healing process [199]. Because of that, the presence of antioxidants is a necessity in the wound healing process. A list of compounds reading antioxidant is shown in Table 2.

#### 3.2.1. Curcumin

Curcumin is mainly extracted from turmeric (*Curcuma longa* L.) and has shown several bioactive properties such as anti-inflammatory, antioxidant, and anti-coagulant [203,204]. Several studies demonstrated curcumin’s wound healing effects as an antioxidant [57,60,82,202]. Phan et al. confirmed that curcumin protects human dermal fibroblasts and epidermal when exposed to hydrogen peroxide and superoxide radicals [60]. Gadekar et al. evaluated the protective potential of curcumin against keratinocytes and fibroblasts in H_2_O_2_-induced injury [82]. Through the antioxidant activity, Bonte et al. also demonstrated that curcumin protects human keratinocytes from xanthine oxidase damage [202]. Mohanty et al. reported the ability of curcumin to reduce ROS and lipid peroxidation, thereby reducing the activation of antioxidant enzymes after wound treatment in rats [57]. The above studies show the impact of Curcumin in the role of an antioxidant in wound healing and its potential in developing methods of using Curcumin in treating wounds.

Despite its excellent biological effects, curcumin has limitations in its therapeutic use because it is virtually insoluble in water leading to instability and poor bioavailability [205].

#### 3.2.2. Quercetin

Quercetin is known as a flavonoid found in many vegetables, fruits, and seeds such as citrus, onion, tea, spices, etc. It is also a famous strong antioxidant and anti-inflammation activities compound [206]. Kant et al. showed that quercetin (0.3%) helps the wound heal the fastest and significantly improves oxidative stress, regulates cytokines and growth factors, and promotes fibroblast proliferation, formation of vessels, and collagen deposition [64]. Mi et al. presented an intensive study evaluating the wound healing effects of Quercetin, which is extracted from Oxytropis falcata Bunge, a traditional Chinese legume distributed in Tibet [63]. This study showed that quercetin-treated wounds had an increase in collagen fiber content and a significant decrease in inflammatory factors (TNF-α, IL-1β and IL-6). In addition, glutathione (GSH) is an antioxidant and an important redox regulator controlling the inflammatory process [207]. Mi et al. also showed that quercetin treatment improved GSH levels suggesting quercetin has a potent antioxidant capacity in skin wounds. In brief, quercetin exhibits an effective wound-healing effect on the skin by enhancing fibroblast migration and proliferation, and inhibiting inflammation through antioxidant activities.

Like most flavonoids, quercetin is poorly soluble in water [208]. This physical limitation affects the application of quercetin in wound treatment. Therefore, further studies on the combination of quercetin are needed to increase its applicability in the future.

#### 3.2.3. Catechin

Catechin is a flavonoid with good antioxidant activity; it plays a beneficial role in physiological activity [160,209]. Baek et al. prepared a PCL/(+)-catechin/gelatin film and evaluated its applicability for wound treatment [200]. The results show that PCL/(+)-catechin/gelatin film prevents harmful factors from the outside, and reduces oxidative stress at the wound effectively to help the wound heal. Zhao et al. confirmed that the EGCG-3-acrylamido phenyl boronic acid-acrylamide (EACPA) hydrogel has antioxidant, antibacterial, antiinflammatory, and proangiogenic effects, and modulates macrophage polarity to accelerate wound healing, also facilitates easy dressing change [65]. This study clearly shows the effect of the antioxidant EACPA on wound healing through the down-regulation of the majority of intracellular ROS in Rosup-stimulated L929 fibroblasts.

Despite having such outstanding activities, catechins are less stable in water. To overcome this problem, several studies were carried out using reducing agents and the formation of micro- and nanoparticles [66,210].

#### 3.2.4. Galic Acid (GA)

Galic acid (GA) is present in almost every plant. It is found in many different parts of plants, such as fruits, leaves, and stems, with powerful properties such as antioxidant, antiinflammation, anticancer, and neuroprotective [211,212,213]. Yang et al. conducted research to evaluate the effects of GA on wound healing in normal and hyperglucidic conditions [61]. This study indicated that GA could protect skin cells from oxidative stress induced by H_2_O_2_ and ROS-induced cytotoxicity. Additionally, GA could upregulate the expression of antioxidant genes such as catalase (CAT), superoxide dismutase 2 (SOD2) and glutathione peroxidase 1 (Gpx1) (Figure 5). Furthermore, GA also accelerates keratinocyte migration during wound healing and activates wound healing factors such as c-Jun N-terminal kinases (JNK), focal adhesion kinases (FAK), and extracellular signal-regulated kinases (Erk). Therefore, this study indicated that GA is a promising antioxidant for wound treatment. However, GA is only soluble in organic solvents, which limits its topical applications on the skin.

#### 3.2.5. Resveratrol (RSV)

Resveratrol is found in more than 70 different plant species and is known for its outstanding medicinal properties such as antioxidant, anticancer, anti-inflammatory, and antibacterial properties [214,215,216,217]. Zhou et al. examined the wound healing ability of resveratrol through the cell and in vivo experiments [149]. Resveratrol protects from H_2_O_2_-induced injury, effectively decreases H_2_O_2_-induced injured cell migration, and effectively suppresses intracellular ROS production by H_2_O_2_ in HUVECs. In vivo tests also confirmed that resveratrol speeds up wound healing, improves skin structure, and reduces inflammation (Figure 6). These effects may be due to resveratrol upregulating Mn-SOD, thereby reducing oxidative damage. On the other hand, Bilgic et al. evaluated the wound healing ability of resveratrol in Wistar albino rats [218]. They showed that the resveratrol-treated wound had a higher neovascularization level than the untreated control group. Furthermore, levels of glutathione peroxidases, enzymes that remove reactive oxygen and nitrogen species from the body, were higher in the resveratrol treatment group. These results suggested that resveratrol affected wound healing through its antioxidant effects.

#### 3.2.6. Naringenin

Naringenin is known as a polyphenol, mainly found in citrus fruits, with outstanding biological properties such as anti-inflammatory, antioxidant, cholesterol-lowering, and anticancer [219,220]. Al-Roujayee et al. evaluated the effect of naringenin in rats for inflammatory responses and oxidative stress caused by thermal burn-induced [201]. The results showed that when the burn was treated with naringenin, the activities of glutathione-S-transferase (GST), superoxide dismutase (SOD), catalase, glutathione peroxidase (GPx), and catalase increased. Thiobarbituric acid reactive substances (TBARS) and glutathione (GSH) levels were also restored on day 7 of treatment. In addition, naringenin was also used to combine with other compounds (e.g., chitosan) to improve wound healing capacity. Akrawi and colleagues showed that a nanoemulsion product containing both naringenin and chitosan significantly increased wound contraction in Wistar rats after 14 days of treatment, and naringenin stimulated antiinflammatory and antioxidant effects (Figure 7) [72]. These results suggest the potential for the treatment of burn wounds of naringenin base on antioxidant activities.

### 3.3. Natural Compounds with Antibacterial Properties

The antibacterial activity of a compound could be ascribed by two mechanisms: inhibition of synthesis of vital components of bacteria or suppression of antibacterial resistance [221]. Natural compounds with antibacterial properties might target mostly the inflammation phase of wound healing (Table 3).

#### 3.3.1. Chitosan and Chitin

The first use of chitosan and chitin (Figure 8) as wound healing accelerators dates back to the research of Prudden et al. [224]. Chitin (poly-*N*-acetyl-d-glucosamine-(1–4)-poly-*N*-acetyl-d-glucosamine) is one of the most prevalent polysaccharides with the largest source from the exoskeleton of marine crustaceans, shrimp, crabs, insects, fungi, and yeasts after cellulose [225]. Chitosan is a copolymer of glucosamine and *N*-acetylglucosamine units connected by 1–4 glucosidic linkages and is the most important chitin derivative.

The secret to the antibacterial capabilities of chitosan is that positively regulating substances make it more susceptible to interacting with negatively charged molecules in bacterial membranes, such as anionic polysaccharides, proteins, and nucleic acids [226,227]. Chitosan has significant advantages in wound treatment due to its biocompatibility, biodegradability, nontoxicity, adsorption properties, and hemostatic qualities [228,229,230]. However, chitosan is insoluble in neutral and alkaline aqueous solutions with pH values greater than 6.5, severely restricting its use [231]. Therefore, chitosan has been integrated into several formulations employing nanoparticles, hydrogel, micelles, hyaluronic/oleic acid-loaded, and glucosylation of the hydrophobic molecule in pre-clinical investigations to improve its bioavailability [53,232,233,234,235].

#### 3.3.2. Honey Bee

Bee products are also natural antibacterial sources widely used in wound healing. Honey from bees has been applied to wound treatment for thousands of years, with the first written recorded between 2600 and 2200 BCE in an ancient Egyptian trauma manual [51,236]. Honey is a concentrated aqueous solution of inverted sugars that contains 40% fructose, 40% glucose, 20% water, enzymes, vitamins and minerals, with a pH of 3.6 [237,238]. Most conventional honey produces hydrogen peroxide by the endogenous enzyme glucoseoxidase, which is responsible for its antibacterial activity. When hydrogen peroxide decomposes, it produces highly reactive free radicals, which react with the bacteria and decimate them [238]. However, several other “non-peroxide” kinds of honey (Ex. Manuka, jelly bush) own antibacterial properties because of the low pH medium and supersaturated sugar level [239]. Especially, Atrott and Henle suggested that Manuka honey has significant levels of methylglyoxal, a unique antibacterial component solely responsible for the special antibacterial effect [52].

#### 3.3.3. Propolis

Propolis was used by ancient Egyptians, Romans, and Persians. Propolis could be obtained from honey bees, tree buds, and other botanical sources (e.g., poplar, willow, elm, alder, birch, beech, etc.) [240,241,242]. Propolis consists of more than 300 chemical compounds such as polyphenols, phenolic aldehydes, amino acids, steroids, etc. [243,244,245]. The most important components in propolis are flavonoids, phenylpropanoids, cinnamic acids and their esters, and glycerides [246,247]. The antibacterial properties of propolis against Gram-positive bacteria also appear mostly due to flavonoids, esters, and aromatic acids found in the resin [246].

#### 3.3.4. Tannins

Along with animal products that have antibacterial activity, many medical plants used in wound healing also show potent antibacterial properties such as tannins. Su et al. reported that tannins extracted from *Entada phaseoloides* (L.) Merr. exhibited the antibacterial property by suppressing protein synthesis, modification of nucleic acid metabolism, prevention of alteration of cell wall formation, modification of cell membrane function, and inhibition of bacterial growth [105].

#### 3.3.5. Allicin

Allicin, the chemical responsible for the strong odor of garlic, is the active ingredient that has been proved in numerous trials to enhance wound healing [127,248,249,250]. Apart from antioxidant activity, allicin also shows an antibacterial effect, and its mode of action has already been researched. The sulfhydryl alteration of bacterial proteins was found to be the mechanism by which allicin manifests its antibacterial activity toward *Staphylococcus aureus* [248,251].

#### 3.3.6. Terpene Esters

Terpene esters could be extracted from bee propolis [252]. Terpene esters demonstrated antibacterial activity toward *Staphylococcus aureus*, as shown in the study of Trusheva and colleagues [252]. The mechanism of the antibacterial activity of terpene esters has not been fully elucidated.

### 3.4. Natural Compounds with Collagen Promotion Properties

Collagen is the protein that is most prevalent in the body. Collagen function in wound healing is to draw fibroblasts and promote the deposition of fresh collagen in the wound bed. The use of collagen dressing technology aids in promoting the formation of new tissues while promoting angiogenesis, autolytic debridement, and re-epithelialization. Hence, the compounds capable of promoting collagen synthesis in the healing process play an important role. A list of compounds with collagen promotion is shown in Table 4.

#### 3.4.1. Saponins

Saponins are glycoside compounds widely found in the plant kingdom. Saponins include various groups and are categorized according to their structure [258]. For instance, Wang et al. reported four novel steroidal saponins, together with two known compounds (i.e., bletilnoside A and 3-*O*-β-d-glucopyranosyl-3-epi-neoruscogenin), were extracted from *Bletilla striata* which is a popular traditional Chinese herb [153]. Numerous biological processes, including hemolysis [259], antibacterial [260,261], antiviral [262], antioxidative [263], antiinflammatory activities [264,265], and collagen promotion [44] can be enhanced by saponin treatment. Yu et al. explored the function of Panax notoginseng saponins (PNS) in encouraging anterior cruciate ligament (ACL) fibroblast migration, proliferation, and expression of fibronectin, collagen I, and collagen III to the healing of an ACL injury. PNS may play an essential role via phosphorylating PI3K, AKT, and ERK [44].

#### 3.4.2. Cryptotanshinone

Cryptotanshinone extracted from *Salvia miltiorrhiza Bge* is a natural accelerated procollagen compound in the wound healing process. Improved angiogenesis and collagen deposition can result from the activity of cryptotanshinone, which reduce leukocyte infiltration, enhance eNOS phosphorylation, boost VEGF and Ang-1 protein production, suppress MMP2 and MMP9 protein expression, and increase fibroblast translation [49].

#### 3.4.3. Artocarpin

There have been claims that the prenylated flavonoid artocarpin, isolated from the plant Artocarpus communis, has anti-inflammatory and anticancer activities [266,267,268,269,270]. Yeh et al. demonstrated that by stimulating the JNK and P38 pathways, Artocarpin boosted collagen formation, proliferation, and migration of human fibroblasts. Artocarpin also enhanced the proliferation of human endothelial cells through the Akt and P38 pathways and human keratinocytes through the ERK and P38 pathways [50].

#### 3.4.4. β-Glucans

β-glucans are glucose polymers, and they can be found in yeast, grains, and fungi. These substances are classified as biological response modifiers [271]. Many studies have demonstrated that particulate and soluble β-glucans improved immune functions with anti-infective, anticancer, and immunomodulatory effects [272,273,274]. β-glucans improve wound healing by enhancing the infiltration of macrophages, which drives tissue granulation, collagen deposition, and re-epithelialization. With excellent stability and resistance to wound proteases, β-glucan-based wound dressings constitute an ideal wound healing agent [275].

#### 3.4.5. Amino Acids and Peptides

Besides the traditional medicinal plants, the sources of natural procollagen compounds containing amino acids and peptides for wound healing also from animals (e.g., bees, mollusks, snail, fish, etc.) are widely reported. For fibroblasts, which need an acidic environment to perform tasks like migrating and organizing collagen, the low pH of honey may help establish and maintain ideal circumstances [276]. Badiu et al. indicated that amino acids from Rapana venosa and Mytilus galloprovincialis enhance dermal and epidermal neoformation to hasten skin wound healing [70]. Indeed, the mechanism insight of these amino acids’ enhancing wound healing effects was proposed to be closely related to differential regulation of macrophage arginine metabolism, in which TGF-β1 may play an essential coregulatory role [277]. In addition, the bioactive peptide extracted from terrestrial snail Cryptozona bistrialis stimulates in vitro migration of NIH/3T3 mouse fibroblast cells. In vivo tests on healthy and diabetic-induced Wistar albino rats also showed that the Crypto-zona bistrialis-peptide was efficient in boosting wound healing [71]. The increased wound contraction is believed to be due to the significant increase in collagen content through the enhanced migration of fibroblasts and epithelial cells to the wound site. However, the extract compounds from animal sources had not shown the exact chemical formula.

## 4. Current Trending Use of Natural Compounds in Wound Healing

The market size for advanced wound care technologies is estimated to be $22 billion by 2030, which will focus on new wound care technologies such as bacterial burden management and biological therapies [278].

The basic understanding of natural compounds and their treatment limitations have been gradually overcome, creating medical products with outstanding features in wound treatment. Currently, the research into medical materials using natural compounds such as gels and films also shows the effectiveness and potential in the future.

Composite dressing shave replaced traditional dressings by combining wound healing drugs such as natural products (chitosan and diazo resin [68]) and growth factors (collagen sponge [73]) to protect the wound from infection and exchange oxygen with the wound [71]. Electrospun nanofiber mats are also a strategy for wound healing. Curcumin has been complexed with nanofiber mats to avoid its limitation (i.e., water insolubility); combining it with an oil layer can increase the bioavailability of curcumin while keeping the wound moist [116].

A hydrogel is also a new approach to biomaterials for wound healing. Hydrogels deliver curcumin, chitosan, and this natural compound released into the wounds. The formulation of this hydrogel not only provides natural healing properties and forms a moist middle layer for the wound. Hydrogels have become a popular new drug/material and a new research area that improves traditional natural compounds in wound treatment [279].

Microneedles are loaded with natural compounds and can penetrate through the dermis layer of the skin. Some studies have reported the superior wound healing ability of microneedles containing manuka honey and green tea extract compared with conventional skin creams [36,67].

Research directions and application of natural compounds to new technology have contributed to speeding up the healing process, solving the limitations of natural compounds, and improving their effectiveness.

## 5. Conclusions

Wound healing is a complex biological process of recovering devitalized cellular structures with four overlapping phases involving hemostasis, inflammation, proliferation, and remodeling. Effective therapies for wound healing using natural products are highly beneficial for patients due to their easy accessibility and low cost. This work proposed a comprehensive review of natural products for wound healing based on bioactivities from plants and animals, providing an overall picture of the chemical origin of natural products to biological wound healing mechanisms. The main four primary bioactivities of natural products, including anti-inflammation, anti-oxidant, antibacterial, and collagen promotion, are utilized to classify and investigate the targeting phases. Data tables containing detail of origin, bioactivity, targeting phase, experimental model, and type of wound were also provided for readers.

## Figures and Tables

**Figure 1 ijms-23-09573-f001:**
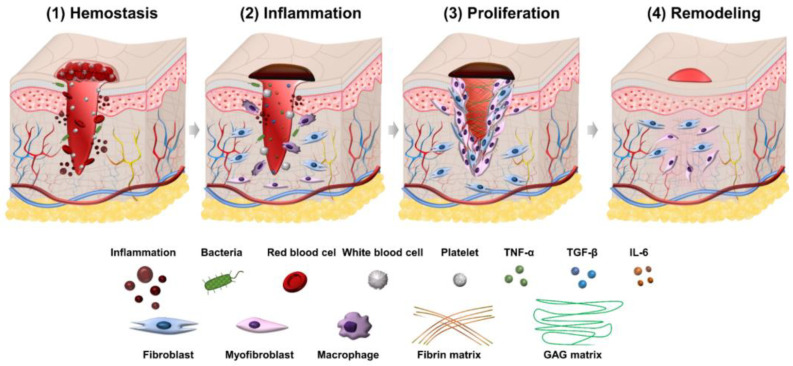
Illustration of four phases in the wound healing process.

**Figure 2 ijms-23-09573-f002:**
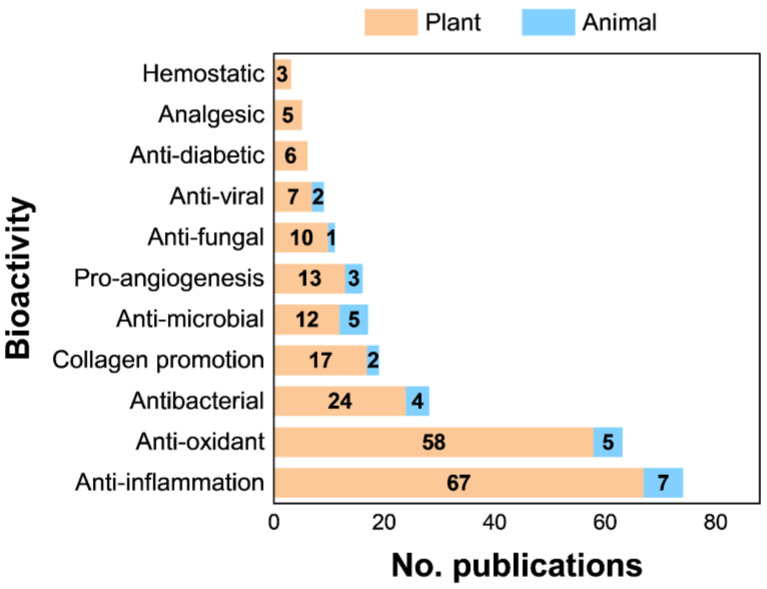
Summary of collected literature based on bioactivities of natural compounds used in wound healing.

**Figure 3 ijms-23-09573-f003:**
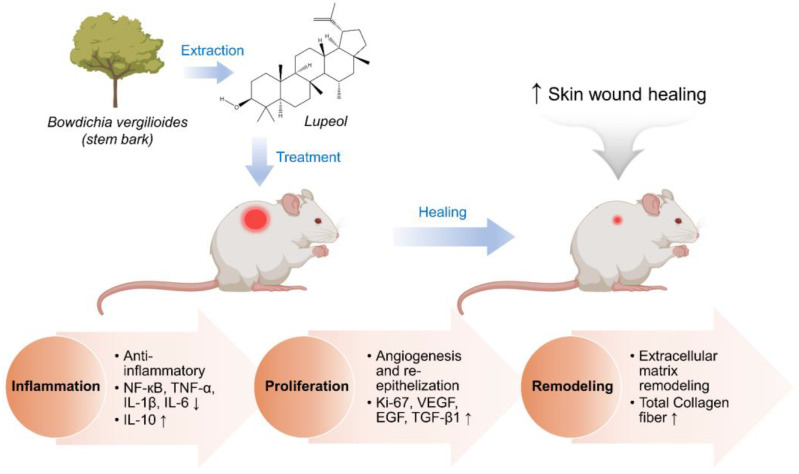
The effect of lupeol cream on wound healing. Up and down arrows mean increasing and decreasing of concentration, respectively. Reproduced with permission from Beserra et al., “From Inflammation to Cutaneous Repair: Topical Applica-tion of Lupeol Improves Skin Wound Healing in Rats by Modulating the Cytokine Levels, NF-κB, Ki-67, Growth Factor Expression, and Distribution of Collagen Fibers”; published by MDPI, 2020 [58].

**Figure 4 ijms-23-09573-f004:**
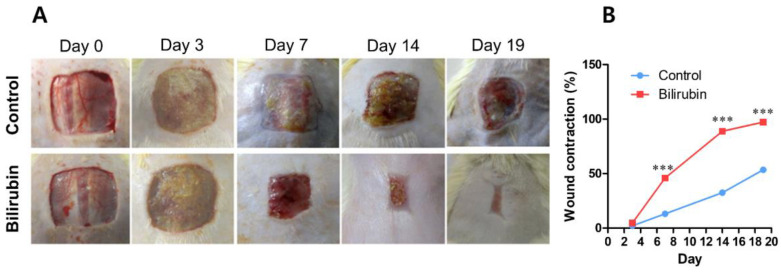
(**A**) Representative images of the wound treated with Bilirubin and (**B**) wound contraction after 19 days. Three asterisks (***) indicates *p*-value < 0.001. Reproduced with permission from Ram et al., “Bilirubin modulated cytokines, growth factors and angiogenesis to improve cutaneous wound healing process in diabetic rats”, published by Elsevier, 2016 [138].

**Figure 5 ijms-23-09573-f005:**
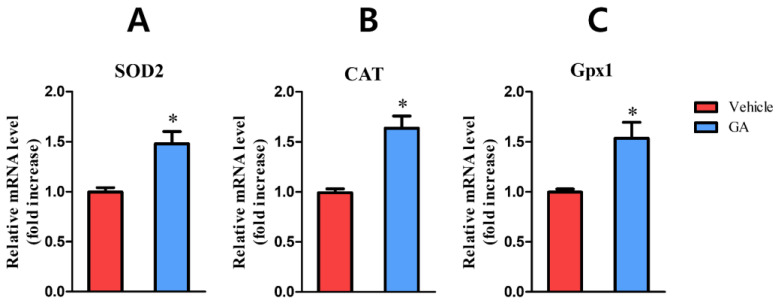
Gallic acid regulates the expression of anti-oxidant genes. (**A**): SOD2, (**B**): CAT, (**C**): Gpx1. Asterisk (*) indicates *p*-value < 0.05. Reproduced with permission from Yang et al., “Gallic Acid Promotes Wound Healing in Normal and Hyperglucidic Conditions”, published by MDPI, 2016 [61].

**Figure 6 ijms-23-09573-f006:**
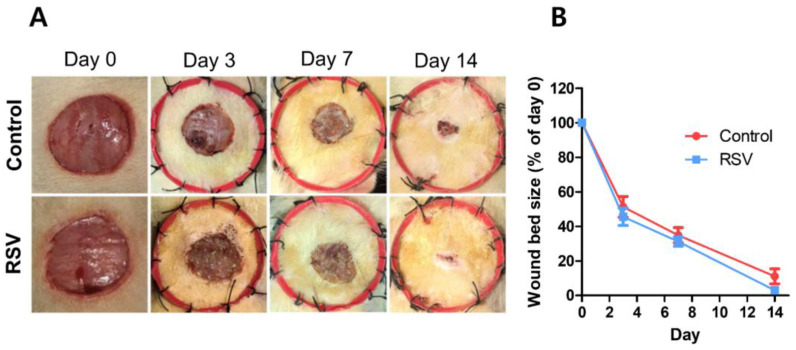
Effect of resveratrol on wound healing. (**A**): Representative images of wound bed size from two groups. (**B**): Quantitation of wound bed sizes. Reproduced with permission from Zhou et al., “Resveratrol accelerates wound healing by attenuating oxidative stress-induced impairment of cell proliferation and migration”, published by Elsevier, 2021 [149].

**Figure 7 ijms-23-09573-f007:**
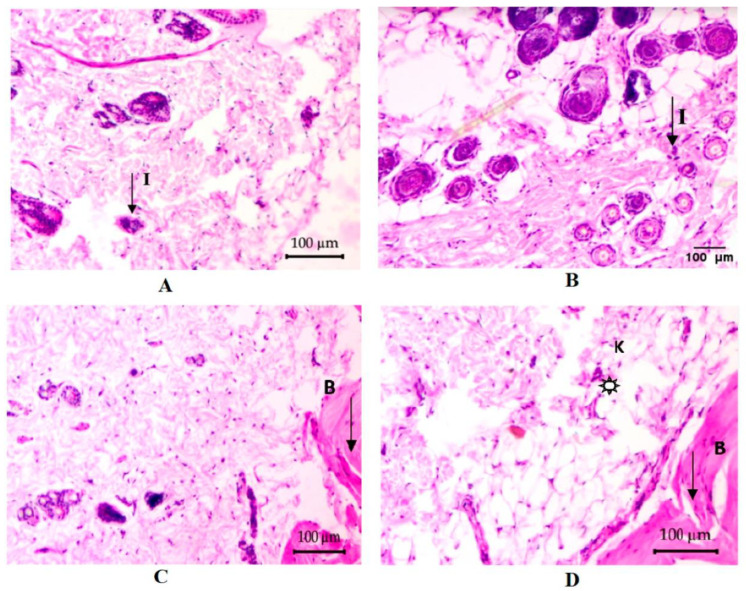
Representative photomicrographs of the rat skin tissues in the control, drug-free chitosan-coated naringenin and chitosan-coated naringenin (CNNE) treated groups of abrasion model in albino Wistar rats. (**A**) Wound area before the treatment at day 0, (**B**) control group at day 14, (**C**) drug-free chitosan-coated naringenin formulation treated group at day 14, and (**D**) CNNE treated group at day 14. (I: inflammatory cells; B: blood vessels; K: keratinization; Star icon: granulated tissue). Reproduced with permission from Akrawi et al., “Development and Optimization of Naringenin-Loaded Chitosan-Coated Nanoemulsion for Topical Therapy in Wound Healing”, published by MDPI, 2020 [72].

**Figure 8 ijms-23-09573-f008:**
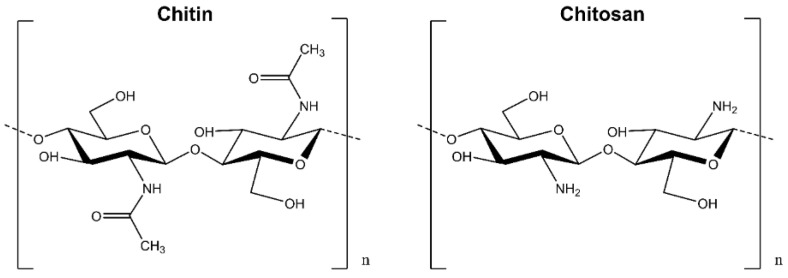
Chemical structure of Chitin and Chitosan.

**Table 1 ijms-23-09573-t001:** Compounds with anti-inflammation.

Compound	Origin	Using Part	Other Bioactivities	Targeting Phase	ExperimentalModel	Type of Wound	Ref.
Asiatic acid	Plant (*Centella asiatica*)	Leaves	Anti-microbial Anti-oxidant Pro-collagen	Inflammation Proliferation Remodeling	Human	Diabetic Burn	[174]
Pinocembrin	Plant	N/A	N/A	Inflammation	HaCaT cell	N/A	[113]
Ursolic acid	Plant (*Hedyotis herbacea*)	N/A	Anti-microbial	Inflammation	Rat	Incision Excision	[175]
Myricetin	Plant (*Tecomaria capensis v. aurea*)	N/A	Anti-oxidant Anti-allergic Analgesic	Inflammation	Rat	Excision	[47]
Myricetin	Plant	N/A	Anti-oxidant	N/A	In vitro	N/A	[59]
Apigenin	Plant	Fruits Beans Tea leaves	Anti-oxidant Pro-angiogenic	Inflammation Proliferation	Rat	Random skin flaps	[109]
Lupeol	Plant (*Bowdichia virgilioides* Kunth)	Stem bark	Anti-oxidant	Inflammation Proliferation Remodeling	Rat	Excision	[58]
Lupeol	Plant (*Bowdichia virgilioides* Kunth)	Stem bark	Anti-oxidant	Inflammation Proliferation	Rat	Excision	[176]
Steroidal glycoside	Plant	N/A	Dermal fibroblast migration activity	Inflammation Proliferation Remodeling	Human dermal fibroblast cells	Human wound	[56]
Verbascoside	Plant (*Plantago subulata*)	Aerial parts	Anti-oxidant Anti-fugal Anti-bacterial Anti-viral	Inflammation Proliferation	L929 fibroblasts RAW 264.7 cells	N/A	[62]
Verbascoside	Plant (*Plantago australis*)	Leaves	Anti-oxidant Healing	Inflammation Proliferation	HaCaT cells Rat	Excision	[177]
Hesperetin	Plant	Citrus species	Anti-microbial Anti-oxidant	Inflammation Proliferation Remodeling	Rat	Excision diabetic foot ulcer	[102]
Hesperetin	Plant	Citrus species	Anti-oxidant Pro-collagen	Inflammation	Rat	Diabetic foot ulcer	[154]
Carophylolide	Plant (*Calophyllum inophyllum* Linn)	Seed	Anti-microbial Anti-coagulant	Inflammation	Mice	Incision	[55]
Artocarpin	Plant (*A.communis.*)	Heartwood	Anti-oxidative, Anti-microbial	Inflammation Proliferation	Mice HUVECs cells	Excision	[50]
Bilirubin	Mammals	Product of heme catabolism	Anti-oxidant	Inflammation Proliferation Remodeling	Rat	Excision	[138]

**Table 2 ijms-23-09573-t002:** Compounds with anti-oxidant.

Compound	Origin	Using Part	Other Bioactivities	Target Phase	Experimental Model	Type of Wound	Ref.
Quercetin	Plant (Oxytropis *falcata* Bunge)	Fruits	Anti-inflammatory Anti-infection	Inflammation Proliferation Remodeling	Mice	Excision	[63]
Resveratrol	Plant	N/A	Anti-inflammatory Anti-bacterial	Inflammation Proliferation	HUVE cells Rat	Burn injury	[149]
Catechin	Plant (Green tea)	N/A	Anti-bacterial Anti-inflammatory Pro-angiogenic	Inflammation	Mice	Chronic diabetic wound	[65]
Catechin	N/A	N/A	N/A	N/A	Mouse NIH/3T3 fibroblast cell	N/A	[200]
Luteolin	Plant	N/A	Anti-inflammatory Anti-allergenic	Inflammation Proliferation	Rat	Excision	[42]
Syringic acid	Plant	Fruits	Anti-inflammatory Anti-microbial Anti-adipogenic	Inflammation Proliferation Remodeling	Rat	Incision diabetic wound	[133]
Metformin	N/A	N/A	Anti-hypoglycemic	Inflammation Proliferation	Mice	Diabetic wounds	[87]
Naringenin	Plant	Citrus fruits	Anti-inflammatory	Proliferation Inflammation	Rat	Thermally-induced skin damage	[201]
Galic acid	Plant	Fruits Leaves Flower	Anti-inflammatory Analgesic	Inflammation Proliferation	HaCaT MEF HF21 cells	Hyperglucidic conditions	[61]
Ferulic acid	Plant (vegetables, cereals, coffee)	Seed Fruits	Anti-inflammatory Antimicrobial	Inflammation Proliferation	Rat	Excision diabetic wounds	[84]
Curcumin	Plant	Turmeric	Anti-inflammatory	Inflammation Proliferation	Rat	Excision	[82]
Curcumin	Plant (*Curcuma longa*)	Turmeric	Anti-inflammatory Anti-infective	Inflammation	Rat	Excision	[57]
Curcumin	Plant	Turmeric	Anti-inflammatory	Inflammation	Human keratinocytes and fibroblasts	H_2_O_2_ condition	[60]
Curcumin	Plant (*Curcuma longa*)	Turmeric	N/A	Inflammation	Human keratinocytes	Hypoxanthine/xanthine oxidase injury	[202]

**Table 3 ijms-23-09573-t003:** Compounds with anti-bacterial.

Compound	Origin	Using Part	Other Bioactivities	Target Phase	Experimental Model	Type of Wound	Ref.
Chitosan	Animal (*Crab*)	Shells	Anti-microbial Anti-inflammation	Inflammation	Diabetic *db/db* mice	Excision wound	[222]
Pinocembrin	Animal (*Bee*)	PropolisHoney	Anti-oxidation Anti-inflammatory Anti-apoptosis	Proliferation	Human Keloid fibroblast Mice	keloid xenograft	[103]
Lupeol	Plant (*Bowdichia virgilioides* Kunth)	Stem bark	Anti-oxidant Antidiabetic	Inflammation Proliferation Remodeling	Rat	Excision	[58]
Hydrogen peroxide	Animal (*Bee*)	Honey	N/A	Inflammation	HaCaT cells	N/A	[223]
Methylglyoxal	Animal (*Bee*)	Honey (Manuka)	N/A	N/A	N/A	N/A	[52]
Tannins	Plant (*E. phaseoloides* (L.) Merr)	N/A	Anti-oxidant Anti-inflammatory	Inflammation Proliferation Remodeling	Rat	Excision	[105]
Arnebin-1	Plant (*Arnebianobilis*)	Root	Anti-fungal	Proliferation	Rat	Excision	[147]
Hydroalcoholic extract	Plant (*Caseariasylvestris* Sw.)	Leaves	Anti-inflammatory Antiseptic	Proliferation	Rodent	Scald burns	[83]
Dichloromethane andhexanoic fractions	Plant (*Calendula officinalis* L.)	Flower	Anti-inflammatory Anti-septic	Inflammation Proliferation	Rat	Excision	[129]
Lawsone	Plant (*Lawsonia Alba* Lam.)	Leaves	Anti-fungal Anti-parasitic Anti-viral	N/A	Rat	Excision Incision	[111]

**Table 4 ijms-23-09573-t004:** Compounds with collagen promotion.

Compound	Origin	Using Part	Other Bioactivities	Target Phase	Experimental Model	Type of Wound	Ref.
Honey	Animal (Bee)	Honey	Anti-bacterial	Proliferation	Rat	Excision	[253]
Calendula officinalis extract	Plant (*Calendula officinalis*)	Flower	Anti-bacterial	Proliferation	Rat	Excision	[98]
Saponins	Plant (*Panax Notoginseng*)	Root Rhizome	Anti-inflammation Anti-oxidant Anti-apoptosis Anti-coagulation	Remodeling	Hypertrophic scar fibroblast	N/A	[165]
Cryptotanshinone	Plant (*Salvia miltiorrhiza* Bge.)	N/A	Anti-inflammatory Anti-oxidative Anti-bacterial	Remodeling	Diabetic mice	Excision	[49]
Bexarotene, Taspine, and 2-hydroxy-1-naphthaldehyde Isonicotinoylhydrazone	Plant (*Daemonorops draco*)	N/A	Anti-bacterial Anti-inflammation	Inflammation Proliferation	THP-1, HaCaT, NIH-3T3 cells	N/A	[35]
Sesamol	Plant	Sesame oil	Anti-inflammatory Anti-oxidant	Inflammation Proliferation	Rat	Diabetic foot ulcer	[86]
Astragaloside IV	Plant (*Astragali Radix*)	N/A	Anti-inflammatory Anti-oxidative	Inflammation Proliferation	Mice	Excision	[108]
Polysaccharide APS2-1	Plant (*Astragalus membranaceus*)	Roots	Anti-inflammatory	Inflammation Proliferation	Mice	Excision	[163]
Aloe vera gel	Plant (*Aloe vera*)	Leaves	Anti-inflammatory Anti-bacterial Anti-viral Anti-fugal	Proliferation	Mouse embryonic fibroblasts	N/A	[254]
Asiaticoside	Plant (*Centella asiatica*)	Aerial parts	Anti-oxidant	Proliferation	Rabbit	Incision	[135]
Gallic acid and quercetin	Plant (*Glycyrrhiza glabra* L.)	Roots	Anti-inflammatory Anti-bacterial Anti-microbial Antioxidant	Inflammation Proliferation	Pig	Excision	[88]
Asiatic acid	Plant (*Centella asiatica*)	Aerial parts	Anti-oxidative	Proliferation	Rat	Wound burn	[141]
β-Glucans	Fungi	N/A	Anti-biotic	Proliferation	Human dermal fibroblasts	N/A	[255]
Alkaloids	Plant (*Evolvulus alsinoides*)	Aerial parts	Anti-bacterial Anti-fugal Anti-oxidant	Proliferation	Rat	Incision	[256]
Asiaticoside and madecassoside	Plant (*Centella asiatica*)	N/A	Anti-oxidant	Proliferation	Rat	Burn injury	[89]
Triterpenes	Plant (*Buddleia scordioides*)	Leaves	N/A	Proliferation	Diabetic rat	Incision Excision	[257]
Deoxyelephantopin	Plant (*Elephantopus scaber*)	Leaves	Anti-inflammatory	Inflammation Proliferation	Rat	Incision	[139]

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
