# Peer review of "A Comprehensive Review of Natural Compounds for Wound Healing: Targeting Bioactivity Perspective"

_ijms, 2022, doi:10.3390/ijms23179573_

Round 1

Reviewer 1 Report

The article is an overview of plant compounds that have a beneficial effect on wound healing. The review describes in detail the mechanisms of healing and the positive effect of plant-derived substances on this process. The review is interesting, structured and well written. I have one comment that may improve this review. Abstract and conclusions look the same. To increase the significance of this article, it would be advisable to rewrite the conclusion.

Author Response

Point 1: “The article is an overview of plant compounds that have a beneficial effect on wound healing. The review describes in detail the mechanisms of healing and the positive effect of plant-derived substances on this process. The review is interesting, structured and well written. I have one comment that may improve this review. Abstract and conclusions look the same. To increase the significance of this article, it would be advisable to rewrite the conclusion.”

Response to the point 1:

Thank you for comments. Based on your suggestion, we rewrote the Abstract and Conclusions to improve the clarity of our manuscript.

 Reviewer 2 Report

Van Anh et al present a comprehensive review on naturally sourced compounds for wound healing applications. Whilst this is a popular review topic, the authors focus on treatments targeting the phases on wound healing, which goes some way to differentiate this review from others.

Unfortunately, the manuscript suffers from a poor level of written English. Whilst generally understandable, a lot of sentences were subject to clumsy sentence structure, which needed a bit of deciphering to interpret. The entire manuscript requires an overall and editing by expert in the field with fluency in the English language. Much of the text also lacks scientific terminology and writing style. Some examples of poorly written sentences and [suggested corrections]:

"Numerous works have been published [that provide] diverse [reviews] of natural compounds for wound healing [applications, which] separated [the] approaches [based on differential] categories, bioactivities, and [modes] of action."

 "In this work, we [provide] a comprehensive review of natural compounds [sourced from both plants and animals that target the different phases of healing to promote wound resolution]."

 "Wounds can happen by accident or medical problems." --> e.g. [Wounds occur as a result of accidental or surgical trauma and from a variety of medical conditions]

 "Wound treatment is done mainly by..." --> e.g. [Wound treatment is mainly performed by...] 

"Ataide and colleagues discussed the natural [activities] of [pro-]wound healing [compounds] and their mode of action..."

 "Closure of acute and chronic wounds is regarded [the] wound healing end point in [the majority of] clinical settings..."

 *Please note these are not the only instances in the manuscript. Proof reading is required throughout.*

 Specific comments:

In Figure 1, presumably the authors mean 'Myofibroblast' instead of 'Myoblast' ?

In the section on the proliferative phase, the authors write, "After migrating into the matrix, fibroblasts change shape, settle down, and begin to proliferate and generate granulation tissue components such as collagen, elastin, and proteoglycans." The use of non-scientific language is evident, but also, this section lacks referral to the important process of activated fibroblast differentiation to myofibroblasts. Other than a brief statement on wound contracture, there is a distinct lack of information provided for this important cell type. 

I believe several plant derived terpene esters, such as those from Euphorbia peplus and the Blushwood berry have shown impressive bioactivities in regulating cells involved in enhancing wound healing. 

Surprisingly, there is an absence of discussion on traditional Chinese medicines (TCM) or other traditional medicines? Although these are typically blends of multiple herbs and herbal derivatives, it provides an interesting perspective on the argument of individualism versus holism, in the context of natural compound treatments.

Table 4 - Leaves or Leaf?

The section 3.4.5. includes discussions about peptides and amino acids, please clarify whether these amino acids are present in the previously discussed compounds, as that would likely allude to their bioactivity and functional mechanisms. If so, perhaps this section should be retitled and briefly discuss the suggested biological and molecular mechanisms that are the current consensus for the enhanced wound healing effects of the peptide/a.a.-containing natural compounds.

Author Response

Point 1: “Unfortunately, the manuscript suffers from a poor level of written English. Whilst generally understandable, a lot of sentences were subject to clumsy sentence structure, which needed a bit of deciphering to interpret. The entire manuscript requires an overall and editing by expert in the field with fluency in the English language. Much of the text also lacks scientific terminology and writing style. Some examples of poorly written sentences.”

Response to the point 1:

Thank you for comments. Based on your suggestion, we sent our manuscript to an English editing service to improve level of written English. Some sentences which were pointed out by the reviewer were rewritten as below:

In the original manuscript (Page 1, Line 16-18):

Numerous works have been published providing a diverse review of natural compounds for wound healing with separated approaches, including categories, bioactivities, and mode of action.

In the revised manuscript (Page 1, Line 16-18):

Numerous published works provided reviews of natural products for wound healing applications, which separated the approaches based on different categories such as characteristics, bioactivities, and modes of action.

In the original manuscript (Page 1, Line 18-19):

In this work, we provided a comprehensive review of natural compounds for wound healing based on targeting phases from both plants and animals.

In the revised manuscript (Page 1, Line 19-20):

In this work, we provide a comprehensive review of natural compounds sourced from both plants and animals that target the different phases of healing to promote wound resolution.

In the original manuscript (Page 1, Line 28):

Wounds can happen by accident or medical problems

In the revised manuscript (Page 1, Line 30):

Wounds occur as a result of accidental or surgical trauma and from a variety of medical conditions.

In the original manuscript (Page 1, Line 36):

Wound treatment is done mainly by...

In the revised manuscript (Page 1, Line 39):

Wound treatment is mainly performed by...

In the original manuscript (Page 2, Line 51):

Ataide and colleagues discussed the natural actives of wound healing and their mode of action

In the revised manuscript (Page 2, Line 54):

Ataide and colleagues discussed the activities of pro-wound healing compounds and their mode of action...

In the original manuscript (Page 4, Line 134-136):

Closure of acute and chronic wounds is regarded to wound healing end point in most clinical settings, yet wounds can continue to undergo remodeling or tissue maturation for months or even years”

In the revised manuscript (Page 4, Line 140):

Closure of acute and chronic wounds is regarded the wound healing end point in the majority of clinical settings...

Point 2: “In Figure 1, presumably the authors mean 'Myofibroblast' instead of 'Myoblast' ?”

Response to the point 2:

Thank you for comments. Based on your suggestion, we changed 'Myoblast' to 'Myofibroblast' in Figure 1 caption.

Point 3: “In the section on the proliferative phase, the authors write, "After migrating into the matrix, fibroblasts change shape, settle down, and begin to proliferate and generate granulation tissue components such as collagen, elastin, and proteoglycans." The use of non-scientific language is evident, but also, this section lacks referral to the important process of activated fibroblast differentiation to myofibroblasts. Other than a brief statement on wound contracture, there is a distinct lack of information provided for this important cell type.”

Response to the point 3:

Thank you for comments. Based on your suggestion, we updated our manuscript as below:

In the revised manuscript (Page 3, Line 100-104):

The formation of granulation tissue is promoted by the modulation of fibroblasts toward myofibroblasts. The myofibroblasts are characterized by the capacity of producing force and of synthesizing extracellular matrix components that allow the contraction of granulation tissue.

Point 4: “I believe several plant derived terpene esters, such as those from Euphorbia peplus and the Blushwood berry have shown impressive bioactivities in regulating cells involved in enhancing wound healing.”

Response to the point 4:

Thank you for comments. Based on your suggestion, we added content of several plant derived terpene esters, such as those from Euphorbia peplus and the Blushwood berry as below:

In the revised manuscript (Page 15, Line 463-466):

Terpene esters could be extracted from bee propolis [252]. Terpene esters demon-strated antibacterial activity toward Staphylococcus aureus as shown in the study of Trusheva and colleagues [252]. The mechanism of antibacterial activity of terpene esters has not been fully elucidated.

Point 5: “Surprisingly, there is an absence of discussion on traditional Chinese medicines (TCM) or other traditional medicines? Although these are typically blends of multiple herbs and herbal derivatives, it provides an interesting perspective on the argument of individualism versus holism, in the context of natural compound treatments.”

Response to the point 5:

Thank you for comments. Based on your suggestion, we updated our manuscript by adding description related to Chinese traditional medicine as below:

In the revised manuscript (Page 10, Line 313-315):

Mi et al presented an intensive study evaluating the wound healing effects of Quercetin, which is extracted from Oxytropis falcata Bunge, a traditional Chinese legume distributed in Tibet [63].

 In the revised manuscript (Page 16-17, Line 478-489):

Saponins are glycoside compounds widely found in the plant kingdom. Saponins include a various group, and are categorized according to their structure [258]. For in-stances, Wang et al. reported four novel steroidal saponins, together with two known compounds (i.e., bletilnoside A and 3-O-β-D-glucopyranosyl-3-epi-neoruscogenin), were extracted from Bletilla striata which is one of popular traditional Chinese herb [153]. Numerous biological processes, including hemolysis [259], antibacterial [260,261], antiviral [262], and antioxidative [263], anti-inflammatory activities [264,265], and collagen promotion [44] can be enhanced by saponin treatment. Yu et al. explored the function of panax notoginseng saponins (PNS) in encourage anterior cru-ciate ligament (ACL) fibroblast migration, proliferation, as well as expression of fibronectin, collagen I, and collagen III to healing of ACL injury. PNS may play an essential role via phosphorylating PI3K, AKT, and ERK [44].

In the revised manuscript (Page 8, Line 270-274):

Pinocembrin is currently used as a traditional Chinese medicine in wound healing [103]. Li and colleagues investigated effects of pinocembrin on skin fibrosis by in vitro and in vivo approaches [103]. The study showed that pinocembrin could significantly reduce bleomycin-induced skin fibrosis and fibrosis-related protein expression of keloid tissues in xenograft mice.”

 Point 6: “Table 4 - Leaves or Leaf?”

Response to the point 6:

Thank you for comments. Based on your suggestion, we updated Table 4 and corrected the English mistakes.

Point 7: “The section 3.4.5. includes discussions about peptides and amino acids, please clarify whether these amino acids are present in the previously discussed compounds, as that would likely allude to their bioactivity and functional mechanisms. If so, perhaps this section should be retitled and briefly discuss the suggested biological and molecular mechanisms that are the current consensus for the enhanced wound healing effects of the peptide/a.a.-containing natural compounds.”

Response to the point 7:

Thank you for comments. Based on your suggestion, we retitled the section “3.4.5. Animal product” into “3.4.5. Amino acids and peptides” and discussed about bioactivities and molecular mechanisms that are the current consensus for the enhanced wound healing effects of the peptide/amino acid-containing natural products as below:

In the revised manuscript (Page 17, Line 513-528):

Besides the traditional medical plant, the sources of natural procollagen com-pounds containing amino acids and peptides for wound healing also from animal (e.g., bee, molluscs, snail, fish, etc.) are widely reported. For fibroblasts, which need an acidic environment to perform tasks like migrating and organizing collagen, the low pH of honey may help establish and maintain ideal circumstances [276]. Badiu et al indicated that amino acids from Rapana venosa and Mytilus galloprovincialis enhance dermal and epidermal neoformation to hasten skin wound healing [70]. Indeed, the mechanism insight of these amino acids enhanced wound healing effect was proposed closed related to differential regulation of macrophage arginine metabolism, in which TGF-β1 may play an essential coregulatory role [277]. In addition, the bioactive pep-tide extracted from terrestrial snail Cryptozona bistrialis stimulates in vitro migration of NIH/3T3 mouse fibroblast cells. In vivo tests on healthy and diabetic-induced Wistar albino rats also showed that the Crypto-zona bistrialis-peptide was efficient in boost-ing wound healing [71]. The increased wound contraction believed due to the significant increase in collagen content through enhanced migration of fibroblasts and epithelial cells to the wound site. However, the extract compounds from animal sources had not shown exactly of chemical formula.